# Combined Use of Non-Destructive Analysis Techniques to Investigate Ancient Bronze Statues

**DOI:** 10.3390/s25061727

**Published:** 2025-03-11

**Authors:** Loredana Luvidi, Annalaura Casanova Municchia, Eleni Konstantakopoulou, Noemi Orazi, Marco Ferretti, Giovanni Caruso

**Affiliations:** 1Institute of Heritage Sciences, Italian National Research Council (ISPC-CNR), 00015 Rome, Italy; loredana.luvidi@cnr.it (L.L.); annalaura.casanovamunicchia@cnr.it (A.C.M.); konstantakopoulou_e@hotmail.com (E.K.); marco.ferretti@cnr.it (M.F.); 2Department of Industrial Engineering, University of Rome “Tor Vergata”, 00133 Rome, Italy; noemi.orazi@uniroma2.it

**Keywords:** pulsed thermography, XRF, Raman spectroscopy, non-destructive analysis, ancient bronze statuary

## Abstract

Non-destructive portable techniques for the analysis of cultural heritage items are essential for enhancing our understanding of these objects and providing valuable information for potential restoration interventions. This paper presents a combined use of pulsed thermography, X-ray fluorescence, and Raman spectroscopy to investigate the ancient bronze “Il Togato”, yielding complementary information concerning the techniques used for creating this artefact and its conservation status. Specifically, thermographic analysis has highlighted the presence of many patches of different size used for emending superficial cast defects, weldings used to connect parts separately cast to the main structure, cracks, and defects located in the bronze thickness. On the other hand, XRF provided information on the composition of the gilding which characterises the statue, and supplied an estimate of its thickness through the use of a stratification model. Additionally, Raman spectroscopy has been applied to identify corrosion products. The experimental results presented in the paper provide a comprehensive knowledge of the bronze under investigation and assess the effectiveness of the portable non-destructive techniques employed in the analysis.

## 1. Introduction

Non-destructive investigation methods are of the utmost importance in the field of cultural heritage. They can provide information useful for both scholars, for better understanding the materials and fabrication techniques, and for restores, for assessing the damage condition of the artefact and for planning a suitable intervention. Among the methods widely employed in cultural heritage, infrared thermography (IRT) has been revealed to be very powerful for the analysis of several typologies of items [1,2], such as ancient books [3], paintings [4,5], mosaics [6], historical buildings [7], and bronze statues [8,9]. In this regard, several methods based on the use of thermography for the study of artistic bronzes are reported in the literature. The first publications concern the study of outdoor bronzes using the passive IRT approach, in order to assess both the thermal stress caused by sun radiation [10,11] and to detect structural inhomogeneities, such as cavities or other internal defects [12]. In recent years, the active IRT approach has also been proposed for the analysis of bronze artefacts. In particular, the pulsed thermography (PT) configuration has been applied to obtain information about the bronze structural elements, such as the presence of internal weldings, and to measure the thermal diffusivity of the bronze alloy, a parameter that is strongly dependent on its composition [13]. In the last decade, thanks to the improvements in both IRT instrumentation and data processing procedures, pulsed IRT has been increasingly used for the detection and characterisation of repairs and cold working, and especially for quantitative evaluations [8].

The success of the PT for this kind of application is based on several reasons, such as its non-destructive nature, the possibility to be carried out in in situ surveys, the capability to perform both qualitative investigation of subsurface features, and the quantitative analysis of the thermal transport properties of constituent materials [14]. Such a technique can also be employed in obtaining an overview of bronze features, which is complementary to what is usually obtained by other imaging techniques, such as X- and γ-ray radiography/tomography [15,16]. Finally, thermography makes it possible to address punctual and compositional techniques, which are used to inspect smaller areas and generally require longer measurement times.

The latter application is considered in this paper, where PT was applied to the investigation of the fragments of a gilt Roman bronze statue, named “Il Togato”, combined with X-ray fluorescence spectroscopy (XRF) and Raman spectroscopy. The statue was found in 1878 in the bed of the Tiber River together with other stylistically and chronologically non-homogeneous archaeological findings. All these finds are related to the Arch of Triumph, built by the Augusti at the beginning of Ponte Sisto, and therefore date back before the commissioners’ time [17]. The statue was made by casting its parts separately using the indirect lost-wax method. They were then welded together.

From a conservation point of view, in addition to the fragmentary nature of the parts, the statue shows remnants of gilding, most of which is now lost [17].

In 1910, the statue underwent a restoration in which the 29 fragments were rejoined with cement in their presumably original position.

In the 80s, thanks to an intervention by the Central Institute for Restoration, the cement was removed and then replaced with a suitable support. This consists of removable metal tubular arms holding the pieces together and not visible from the outside, as shown in Figure 1 [18]. The artefact, which is 2 m high, is now on display at the Roman National Museum.

The present research combines three non-invasive techniques that focus on different aspects of the artefact under investigation. Namely, pulsed thermography focuses on material discontinuities, all related to the manufacturing technique and the structural damage that the artefact has undergone over time, as mentioned above. Raman spectroscopy identifies the corrosion products, which is essential for building the layered model used later in the XRF simulations. Finally, X-ray fluorescence was used to study the elemental composition of the alloys and the gilding technique. This study shows that the combination of non-invasive methods provides a reasonably complete understanding of the technical history of the gilded statue, while remaining closely within the limits of non-invasiveness.

PT was here used to analyse some of the areas of the statue, previously selected by visual investigation, affected by casting defects and restorations. In general, infrared thermography is a method widely used in the field of cultural heritage [19]. The technique can be successfully applied to both optically semi-transparent artefacts [20], like library materials and paintings, and optically opaque artworks, such as bronze statues [8,9]. In the so-called pulsed configuration, the infrared radiation emitted by the sample after heating with a visible light pulse (VIS) is analysed. In particular, the radiation is recorded by an infrared camera, operating in the 3–5 μm spectral range, which provides a sequence of temperature maps called thermograms. These images show elements at different depths in the sample. In the case of optically opaque samples, the excitation light pulse is absorbed in a very thin layer beyond the illuminated surface and converted into heat. The infrared emission detected by the camera is contributed by only the sample surface, since the emission of the inner layers is totally absorbed inside the sample, and is essentially proportional to the sample surface temperature. The heat generated in correspondence of the illuminated surface diffuses inside the sample volume. In the presence of inhomogeneous elements located under the illuminated surface, the heat diffusion process is no longer uniform and, as a result, the corresponding areas of the surface have a different temperature than the surroundings, thus appearing in the thermographic image and revealing the inhomogeneous element.

The technique is quite powerful for investigating discontinuities typical of bronze statuary, such as patches, inserts, repairs, welds, internal cracks, and casting flaws. In order to also acquire quantitative information from the measurements, a mathematical model for describing PT in optically opaque items has been employed. It provides numerical results to be compared with experimental ones, for a quantitative evaluation of several parameters such as bronze thermal diffusivity, bronze thickness, and thermal contact resistance of patches applied to emend the casting defects. This latter information is useful in understanding how well the patches originally used to repair surface casting defects adhere to the underlying substrate. The application of the PT to the statue “Il Togato” has revealed, besides the presence of the many patches used to emend the surface casting defects, weldings used to connect parts separately cast, core pins required by the lost wax casting procedure, and some faults laying inside the structure thickness. Several examples are reported in the paper to highlight the effectiveness of the PT technique in bronze statuary analysis.

When dealing with a bronze statue, it is important to analyse its surface, which can undergo changes over time, forming a corrosion layer of varying thickness [21]. The formation and stability of this corrosion layer depend on several factors, such as pH, environmental and conservation conditions, material composition, and the manufacturing process of the artefact [22]. Thus, identifying and characterising the corrosion product layer is crucial for understanding the preservation status of the artwork. Moreover, the bronze under investigation is characterised by the presence of a gilding which is still partially present. Studying its composition can be helpful in understanding how this thin gold layer has been applied to the bronze support. In order to study these surface characteristics, two specific portable techniques have been used in this work, namely, XRF and Raman spectroscopy.

XRF is a well-established technique that provides information on the elemental composition of the alloy, fundamental data for studies of provenance, production techniques, and authenticity of artefacts [23]. XRF analysis also makes it possible to study the gilding of the statue, distinguishing the gilding method used, the thickness, and the possible presence of a corrosion layer between the gilding and the casting alloy. In particular, using PyMca software [24], it is possible to analyse the fluorescent response of the material by modelling the type of corrosion through the definition of possible multilayers [25]. This approach, based on the a priori definition of a model, which is then verified and corrected iteratively by matching the simulated spectrum with the experimental one, made it possible to configure three possible layers, ranging from the outermost one consisting of the gilding to the intermediate layer of the corrosion products, up to the bronze alloy substrate.

Raman spectroscopy has gained widespread use in recent years, particularly in the study of both natural and artificial patinas, especially on copper and bronze surfaces [26,27]. Recently, this technique has also been applied in authentication and provenance studies, based on the recognition of natural and artificial patinas [28]. A significant advantage of Raman spectroscopy over techniques such as electron microscopy (SEM-EDS) or elemental analysis (e.g., XRF), which are limited to detecting only metallic elements, is its ability to identify functional groups and the nature of the formed products, such as sulfates, oxide, carbonates, or chlorides [21]. Furthermore, the ability to acquire spectra directly in situ, using a microscope coupled to a micro-Raman system, allows for the analysis of the corrosion layer in areas previously examined under the microscope, observing optical morphologies.

## 2. Materials and Methods

In this section, the experimental setup relevant to the three non-invasive techniques employed to study the bronze statue are described. The mathematical model of PT applied to the case of optically opaque samples, used in the foregoing to get quantitative information from the measurements, is also briefly recalled. Finally, the processing and analysis of XRF data using PyMca software (software: version 5.9.2) to construct and verify the layer model is described.

### 2.1. The PT Mathematical Model

A tridimensional mathematical model describing the PT applied to the study of a metallic specimen is here recalled [29]. The specimen is composed of a substrate including an insertion, as depicted in Figure 2, where a cartesian frame is introduced as shown.

The model is used to get quantitative information from the PT measurements by comparing the experimental results with the theoretical ones. In particular, it can be useful to estimate the thickness and the thermal interface resistance of the patches used to amend the cast defects after the original casting process. This parameter is closely related to adhesion quality of the insertion on the substrate. The specimen has a thickness *H* and its domain is denoted with Ω. It hosts an insertion of thickness h whose domain is denoted with Λ. The specimen is illuminated by a pulse of visible light (VIS) on the front surface *S* with constant spatial density *I*, and IR indicates the consequent infrared emission from *S*. The interface between insertion and substrate is denoted with *G*, whereas the lateral and rear surface of the substrate is denoted with Σ. The field equations inside the specimen read as:(1)$$\rho_{s}c_{s}\frac{\partial T_{s}}{\partial t}-k_{s}∆T_{s}=0\;\;\;in\;\;\Omega ,\;\;\;\;\;\;\;\rho_{i}c_{i}\frac{\partial T_{i}}{\partial t}-k_{i}∆T_{i}=0\;\;\;\;\;in\;\;\Lambda \;,$$
where *T* is the temperature, *t* is the time, and ρ, c, and k are, respectively, the mass density, the specific heat, and the heat conductivity, with the subscripts *s* and *i* indicating, respectively, the substrate and the insertion.

The equations (1) are equipped with the following boundary conditions:(2)$$k_{s,i}\frac{\partial T_{s,i}}{\partial z}=I\left(t\right)\;\;\;on\;S\;k_{i}\nabla T_{i}\cdot n_{i}=C\left(T_{s}-T_{i}\right)\;\;\;on\;G\;-k_{s}\nabla T_{s}\cdot n_{s}=0\;\;\;\;\;\;\;on\;\Sigma ,$$
where *I* has a step-like profile in time with width equal to the flash duration, *C* is the interface thermal conductivity, and *n_s,i_* denotes the normal to the domain *s*, *i* boundary interface, respectively, pointing outward. Heat losses can be easily accounted for by suitably modifying Equation (2)_3_, for example, considering a flux due to heat convection. However, in the case of metallic specimens, characterised by quite high values of thermal diffusivity, heat losses do not significatively affect the time response in the interval 0–0.5 s. A variational formulation of the proposed model, not reported here for the sake of brevity, is derived and used to supply a finite element formulation, useful to perform the numerical simulations. A program in Matlab R2022a language has been developed based on that finite element formulation, employing a mesh of prisms with triangular bases for the volumes and triangles for the surfaces, with linear shape functions to interpolate the unknown temperature inside each element in terms of its nodal values. The standard isoparametric map was adopted to compute the integrals in the variational formulation and for assembling the mass and stiffness matrices [30]. Finally, the Crank Nicolson method was employed to perform the time integration.

Once the unknown temperature *T* has been computed both inside the substrate and the insertion, the IR radiation detected by the camera is just proportional to the temperature evaluated at the front surface *S*.

The prosed tridimensional model has been validated both in the case of optically opaque samples (copper and bronze) [29], and in the case of optically semi-transparent samples (paper) [20], by comparing the numerical results supplied by the model to experimental results obtained on test specimens.

### 2.2. Experimental Setup

In this section, the experimental setups relevant to the three techniques employed to analyse the bronze “Il Togato” are described.

#### 2.2.1. Pulsed Thermography

The Togato was heated by means of two flash lamps, each one with a maximum electrical power of 3 kW, oriented at 45° to the surface of the bronze. The light emitted by the lamps was filtered to remove the IR spectral component and spurious contributions to the PT signal originating from reflections of the IR radiation on the sample surface [31]. The detection of the emitted IR radiation was obtained using a FLIR6800sc IR camera, (640 × 512 pixels, InSb focal plane array, pitch 25 μm wavelength range 3–5 μm, NETD < 20 mK at 30 °C, scan rate at the maximum spatial resolution 520 Hz), FLIR, Niceville, FL, USA. The described PT setup is schematically depicted in Figure 3. Research Studio^©^ v.3 software was used to process the IR images recorded at a frame rate of 120 Hz. In order to improve the contrast of the recorded thermograms, the frame obtained just before the heating pulse was subtracted from all the frames of the sequence after the flash.

The effect of a non-uniform emissivity of the bronze surface can be compensated by dividing pixel by pixel the signal in all the thermograms after the flash by the signal in the first thermogram just after the light pulse. This was not done in the experiments reported in the foregoing after having verified that this effect is negligible for the scope of the present study.

#### 2.2.2. X-Ray Fluorescence (XRF)

XRF is a well-established technique that has been used to measure the elemental composition of heritage materials for at least 60 years [23]. For the present work, it is useful to remember that the results are significantly affected by the layers of corrosion usually present on the surface of ancient metal artefacts. However, the careful selection of the measurement points and the repetition of the measurements to work with clusters rather than single data points, together with its intrinsic non-invasiveness, make this technique an excellent tool for detecting small compositional differences between similar materials. In addition to providing a reasonably accurate estimate of the composition of the bronze, this research has been used to investigate the gilding in terms of:(a)The method of gilding (leaf or fire gilding);(b)The thickness;(c)The possible presence of a corrosion layer between the gilding and the substrate.

In addition, the possibility of accessing the inner surface of the casting made it possible to remove surface corrosion products and take some measurements on uncorroded bronze. This provided reference compositional values to be used in layered models.

The XRF instrument F-70, specifically designed and assembled at CNR-ISPC by one of the authors (Marco Ferretti, Rome, Italy), for measurements on copper-based artefacts [32], is equipped with a transmission X-ray source and a 20 mm^2^ silicon drift detector (SDD). Further features and the experimental conditions of the system are shown in Table 1.

Under these experimental conditions, the detection limits for the minor elements that are discriminators to distinguish the different alloys are a few tens of mg/kg for Ag, Sn, and Sb, and a few hundreds of mg/kg for Zn, As, Au, Hg, and Pb. Spectra were quantified by the PyMca fundamental parameters (FP) software [24], an open-source software (PyMca version 5.9.2). Calibration was performed using the CHARM set standards [33].

A total of 55 measurement points were selected for analysis, selecting areas of the artefact to study the gilding and the corrosion layer, as well as some internal areas for an accurate measurement of the alloy composition.

In addition to quantification, PyMca was used to simulate the fluorescence behaviour of different layer structures. The procedure consists of hypothesising a layered structure consisting of a gilding with a given composition, an intermediate corrosion layer with a given composition, and a substrate with a composition derived from the fluorescent response of the abraded surfaces as discussed above. This hypothetical structure is then compared with the experimental spectrum. The thickness and composition values of the model are iteratively adjusted to obtain a good match between the modelled and experimental spectra, thus producing a set of geometric and compositional features of the gilding compatible with the experiments.

#### 2.2.3. Raman Spectroscopy

In order to characterise some areas of corrosion highlighted by the XRF investigations, Raman analysis was performed. Raman measurements were conducted using a portable “Raman” spectrometer (B&W Tek, Newark, DE, USA), equipped with a laser source at 532 nm (nominal output power 50 mW). The collecting optics of the system consist of a head with an objective input (for this investigation, 20× and 40× objectives were used), fitted with an optical camera and mounted on an extendable arm on a tripod. The backscattered light is dispersed by an 1800 lines/mm grating, and the Raman signal is detected by a silicon CCD sensor thermoelectrically cooled to −5 °C. The nominal spectral resolution is approximately 5 cm^−1^.

All measurements were collected after an automatic subtraction, referred to as “dark”, to remove any interference from the instrument, substrate, or external light sources in the spectrum. Spectral acquisitions (3 accumulations, 30 s each, in the range 100–4000 cm^−1^) were performed with 20×, 40×, and with a probe that includes a fiber optic interface. The spot size is 85 μm at a working distance of 5.90 mm. The laser power and collection times were varied for each examined area to optimise the signal while preventing sample degradation. Origin 8.5 software was used for spectral processing, including baseline correction to remove background fluorescence.

## 3. Results and Discussion

In this section, the experimental results relevant to the bronze statue under investigation are reported and discussed.

### 3.1. Pulsed Thermography

The use of PT on the bronze statue “Il Togato” has revealed the presence of many patches of different size, used for emending surface casting defects. Many of these patches are hardly visible at a visual inspection since they have been carefully polished after being applied, but they clearly appear in the IR image since the interface between patches and substrate behaves as a discontinuity for the heat diffusion process along the bronze thickness. An example of a series of patches located on the statue drapery is reported in Figure 4, where a visible light image of the drapery is compared with the corresponding IR image. It can be noticed that some patches appear brighter than others in the thermogram, this difference increasing with the time delay after the excitation flash. This occurrence reveals that their mechanical adhesion to the support is weaker, implying a larger thermal resistance at the interface between patch and support. To this end, in Figure 5, the IR signal revealed by the camera in correspondence of the centre of two patches, indicated, respectively, with a red and blue arrow in Figure 4, exhibiting a different level of adhesion is reported versus time in double logarithmic scale.

Both the curves exhibit a straight behaviour just after the flash pulse, with a slope equal to −0.5 in the double logarithmic scale. This behaviour characterises the heat diffusion process generated by a light pulse inside a homogeneous material. After some time, the curves exhibit a change of slope. The time at which this change occurs depends on the patch thickness. It physically represents the time required for the heat generated at the patch surface to diffuse reaching the interface between patch and substrate. After the slope change, a plateau behaviour follows. The width of the plateau is closely related to the contact thermal conductance at the interface: the smaller the conductance, the wider the plateau. By comparing the two experimental curves in the figure, it clearly appears that in the case of better adhesion to the support, i.e., higher interface thermal conductance, the plateau is shorter, whereas in the case of worse adhesion, the plateau is wider. After the plateau behaviour, the curves bend downward and start to decrease again. This last portion of the curves corresponds to the diffusion of the heat beyond the interface inside the substrate [29].

A quantitative analysis can be performed by using the mathematical model previously described, through a curve fit of the two experimental curves. To this end, the model parameter values reported in Table 2 have been used for the numerical simulations. It is here remarked that the value assumed for *I* is not crucial for the comparison with the experimental results, since the mathematical model is linear and then the computed IR emission can be suitably rescaled to match the corresponding experimental curve.

The patch thickness together with its interface thermal conductance have been assumed as fitting parameters, to be estimated by a curve fitting procedure of the experimental curve with the corresponding curve supplied by the mathematical model, by minimizing the least square error between numerical and experimental data using a built-in Matlab routine.

In the two panels of Figure 6, the best fit numerical curves have been reported together with the experimental curves. The agreement between numerical and experimental results is quite satisfying. The estimated value of the patch thickness is 1.2 mm for both the patches, and the interface thermal conductances turned out to be 8000 W/m^2^K and 3500 W/m^2^K, respectively, for the case of better adhesion and worse adhesion. These estimated values are typical of inserts mechanically applied to the support, as it has been assessed in Ref. [24], where suitable laboratory tests have been conducted.

Figure 7 shows the thermogram of the hand of the statue together with the photograph of the same area. In the thermogram taken 0.2 s after the excitation light pulse, a bright ring is clearly visible, corresponding to a welding procedure used to join the finger, which was separately cast, to the rest of the hand.

In Figure 8a, detail of a patch applied on a portion of the drapery is visible in the thermogram, recorded 0.2 s after the excitation light pulse. It is interesting to note the two bright spots that appear in the patch image, corresponding to a deep defect lying under the patch, that was repaired by the application of the patch.

In Figure 9, another portion of the hand has been examined and an internal crack in the bronze clearly appears in the thermogram recorded soon after the excitation light pulse, not visible in the corresponding photograph of the same area.

### 3.2. XRF and Raman Spectroscopy

The composition of the Togato alloy and one of the welds was determined by XRF on a few internal areas of the statue where it was possible to abrade the surface to obtain a measurement unaffected by corrosion products. The average values are given in Table 3.

XRF measurements on the outer surface of the statue in areas where gilding is present revealed the presence of gold with mercury below the detection limits (Table 4). The presence of mercury below the detection limit is characteristic of a ‘pure’ gilding technique, and not only rules out fire gilding, but also the hybrid method using mercury as an adhesive for the gold. Furthermore, the high lead content of the Togato bronze alloy is incompatible with fire gilding, which would be stained by lead during application [34]. Significant and straightforward evidence of the gilding can be obtained by considering only the clean areas of the peaks. Figure 9 plots the net peak areas AuL vs. SnK, AuL vs. PbL, and AuL vs. AgK for all 55 measurements independently of their surface conditions.

The net values of the peak areas allowed for a qualitative reconstruction of the stratigraphy of the different layers, starting from the surface.

As shown in Figure 10a, the Au-Sn diagram reveals low tin (Sn) values in the gilding (yellow dots). The tin comes exclusively from the bronze substrate, and the gilding attenuates its fluorescent radiation. In contrast, Pb appears to be less affected by gold plating, suggesting that there is a Pb-rich layer between the substrate and the gold plating, and that the Pb fluorescence signal from this layer is even higher than that of the substrate (Figure 10b).

Furthermore, images captured with the Dinolite microscope clearly identify the dark red patina (Figure 11) as being located beneath the gilding and above the alloy. In this area, the acquired Raman spectrum showed a peak at 1056 cm^−1^, attributed to the presence of carbonate ions in lead carbonates. These data are consistent with the hypothesis that the intermediate corrosion layer is a mixture of cuprite and lead carbonates. Finally, the gold-silver diagram (Figure 11b) highlights a strong correlation between the two elements, indicating that the detected silver is mainly contained in the alloy of the gold leaf.

With all this in mind, we could build the 3-layer model used in PyMca (version 5.9.2) to estimate the composition and thickness of the gilding and intermediate corrosion layer.

Figure 12 shows the proposed layered model for the analysed bronze statue and the average values of composition and thickness: (1) the gilding layer consists of 88% gold (Au), 4% silver (Ag), and 8% copper (Cu), with a total thickness of 2 µm; (2) the underlying layer of corrosion products (copper oxides ~40% and lead carbonates ~60%) with a total thickness of 20 µm; and (3) the bronze substrate, consisting of 60.0% Cu, 15.0% Sn, 22.8% Pb, and 1.43% iron (Fe). The detailed results are shown in Table 4.

## 4. Conclusions

Various natural science-based methods are available for heritage investigation. Such methods can be more or less accurate and more or less invasive. In most cases, high-accuracy methods require sampling, and non-invasive methods are affected by the unpredictability and inhomogeneity of heritage materials. The main advantage of this approach, which integrates different analytical results, lies in the ability to identify the areas of greatest interest, which can be subsequently examined in depth through a collection of samples for semi-destructive analysis.

This paper aims to explore a rigorously non-invasive approach to the investigation of the Roman gilded bronze statue known as “Il Togato” and discusses the relationship between the outcomes of the research and relevant aspects of the technical history of the Togato.

This work also forms part of the study that the authors have been carrying out in recent years on the ancient statuary in the National Museums of Rome, in order to obtain comparative data on the great Roman bronze masterpieces [8,9,29,35].

The thermographic investigation shows the details of the manufacturing method consistent with the technology of that time, namely, the casting of the artefact in separate sections and joining them by welding. The casting defects visible on the surface were repaired by the mechanical application of patches.

By using a suitable mathematical model, it was possible to perform a quantitative analysis by estimating the patch thickness and the interface thermal resistance. The latter parameter is closely related to the adhesion degree of the patch to the bronze substrate and makes it possible to distinguish patches (high interface thermal resistance) from cast-on repairs (low interface thermal resistance) [29]. Finally, PT produced high-contrast images of the gilded areas, facilitating their mapping. Concerning the limitations of PT, it is here remarked that such a technique can only reveal features located a few millimeters under the bronze surface. In order to study inner elements (i.e., scaffolding, etc.), X-ray radiography/tomography should be employed. The optimal approach is the integrated use of both the techniques to obtain complementary information.

The composition and structure of the surface layers were studied by quantifying and modelling XRF spectra. It was found that the experimental results were consistent with a 3-layers model in which the outermost layer is a gold leaf (other techniques other than “pure” leaf gilding can be excluded), with an average composition and thickness of 88% Au, 4% Ag, and 8% Cu, and 2 μm, respectively [24,25,36]. The absence of mercury or its presence below the detection limit in the outermost layer indicates that the gilding technique is based on the application of gold leaves, probably fixed to the substrate with an adhesive [34,37,38]. However, with our techniques, it was not possible to obtain information on the possible presence of the adhesive. The innermost layer is the bronze substrate whose average composition is 60.5% Cu, 15% Sn, 23% Pb, and 1.5% Fe. It should be noted that such a lead-rich bronze is not suitable for gilding techniques using mercury.

These data are comparable with those of the equestrian statue of Marcus Aurelius [39], characterised by a leaf gilding with thicknesses varying between 3 and 9 μm and a high content of alloying elements. Between the gold leaf and the substrate, the model assumes an intermediate layer consisting of corrosion products of copper and lead, also observed under the optical microscope. A composition compatible with the experiments is 60% PbCO_3_, 40% Cu_2_O, with a thickness of 20 μm. In this layer, the XRF measurements detect a higher amount of lead than that present in the alloy. However, it should be pointed that the Raman band at 1056 cm^−1^ alone does not allow us to distinguish the different forms of lead carbonate (cerussite, hydrocerussite, or plumbonacrite) and to exclude their co-presence. However, these alteration forms are usually white/grey in colour, so the red colouration observed could be due to the presence of copper oxide (cuprite).

Thanks to the interesting results obtained in this completely non-invasive study, we intend to continue the study of the gilding of ancient bronzes, selecting other statues in order to characterise the different gilding techniques used in ancient times to embellish monumental statues. Future research will be based, on the one hand, on the implementation of a database of thermographic features of large bronzes and, on the other hand, on the application of modelling to the study of gilding through the processing of XRF data.

## Figures and Tables

**Figure 1 sensors-25-01727-f001:**
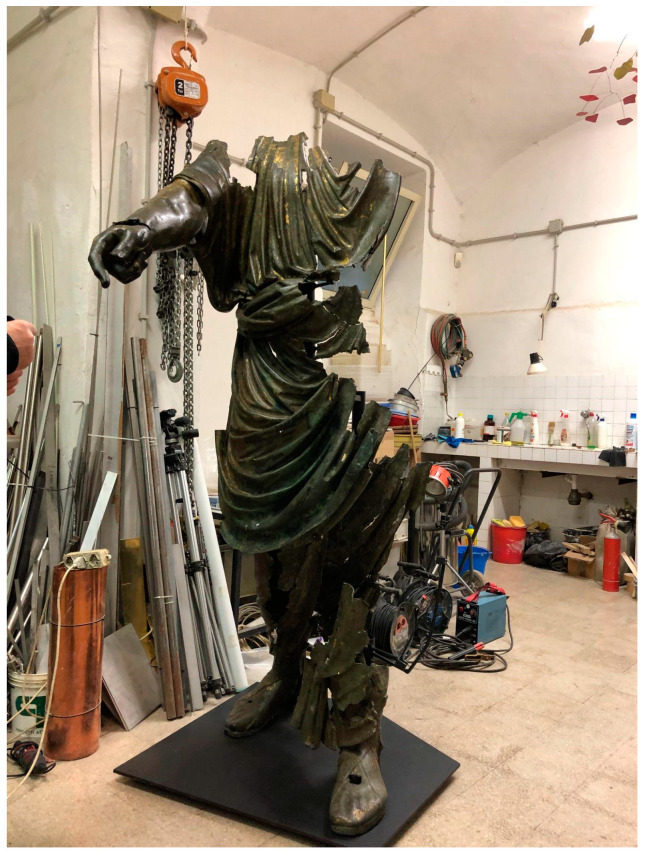
The bronze statue “Il Togato” during a restoration session.

**Figure 2 sensors-25-01727-f002:**
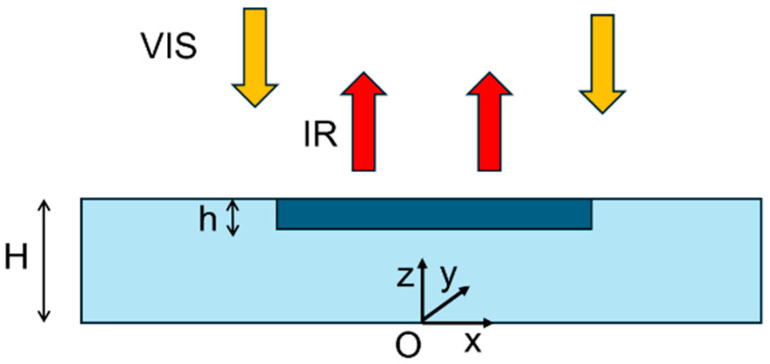
Schematic representation of the specimen used for developing the mathematical model.

**Figure 3 sensors-25-01727-f003:**
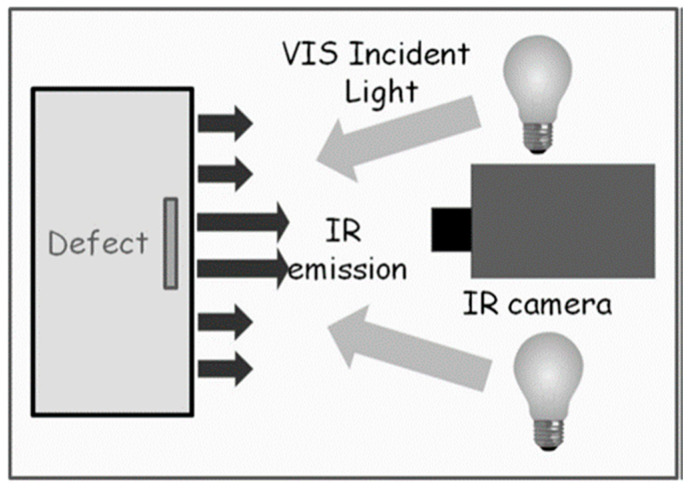
Schematic representation of the PT setup.

**Figure 4 sensors-25-01727-f004:**
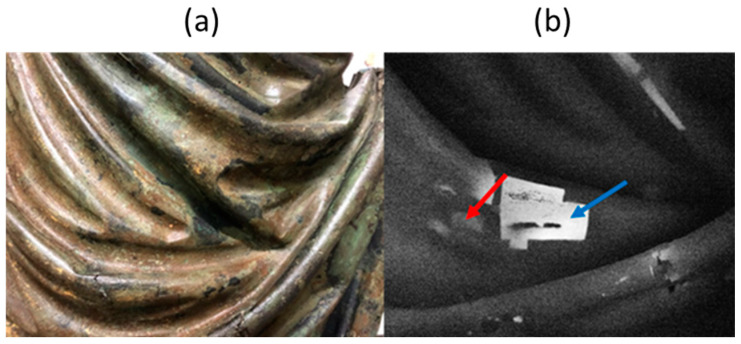
Patches used to emend surface casting defects on the drapery. Photograph (**a**) and thermogram (**b**) recorded 0.2 s after the light flash. Red and blue arrows indicate, respectively, a patch with a good adhesion and one with a bad adhesion to the support.

**Figure 5 sensors-25-01727-f005:**
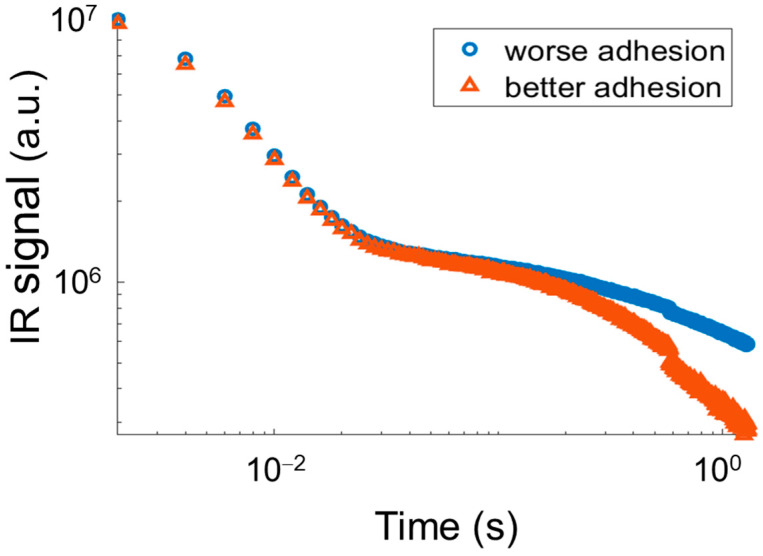
Experimental IR signal versus time in correspondence of the centre of two different patches exhibiting a different adhesion degree to the support. The red curve corresponds to a better adhesion than the case of the blue curve.

**Figure 6 sensors-25-01727-f006:**
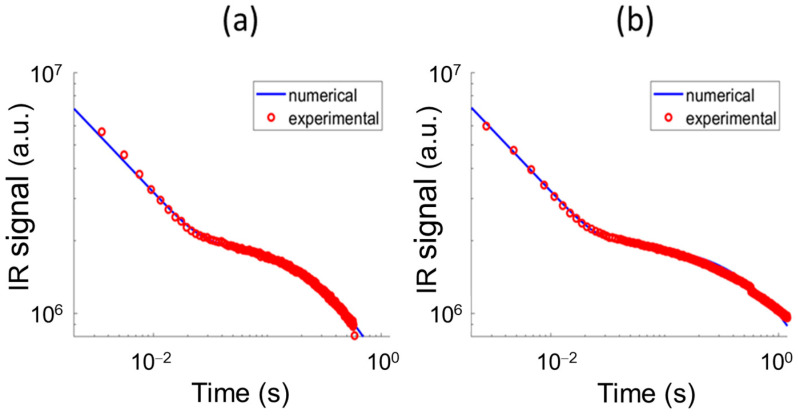
Curve fitting of the two experimental curves shown in Figure 5. The mathematical model previously described was used to compute the theoretical curves, with the parameter values reported in Table 2. The fitting parameters are the patch thickness and its interface thermal conductance. Panel (**a**): better adhesion case, *h* = 1.2 mm, *C* = 8000 W/m^2^K, goodness-of-fit index R^2^ = 0.86; panel (**b**): worse adhesion case, *h* = 1.2 mm, *C* = 3500 W/m^2^K, goodness-of-fit index R^2^ = 0.86.

**Figure 7 sensors-25-01727-f007:**
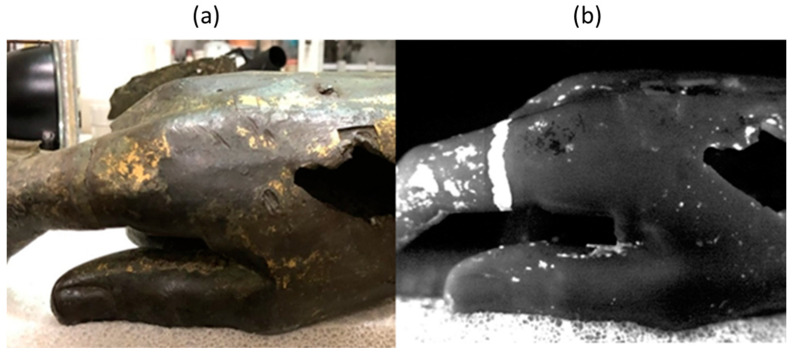
Welding ring for connecting the index finger to the hand. Photograph (**a**) and thermogram (**b**) recorded at 0.2 s after the light flash.

**Figure 8 sensors-25-01727-f008:**
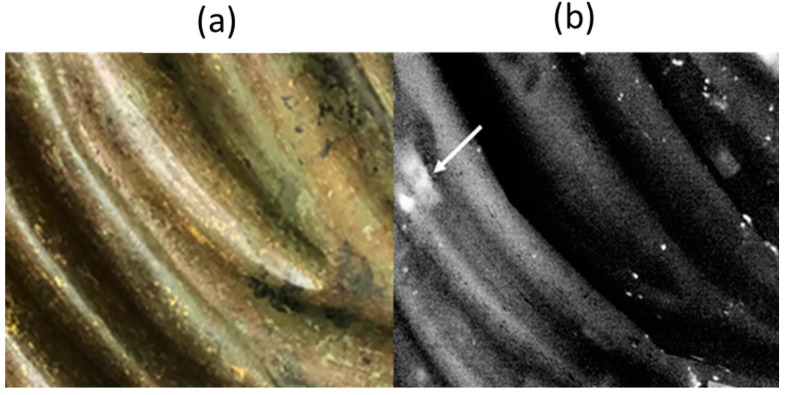
IR image of a patch where the arrow indicates a deep defect located under the patch is clearly visible. Photograph (**a**) and thermogram (**b**) recorded at 0.2 s after the light flash.

**Figure 9 sensors-25-01727-f009:**
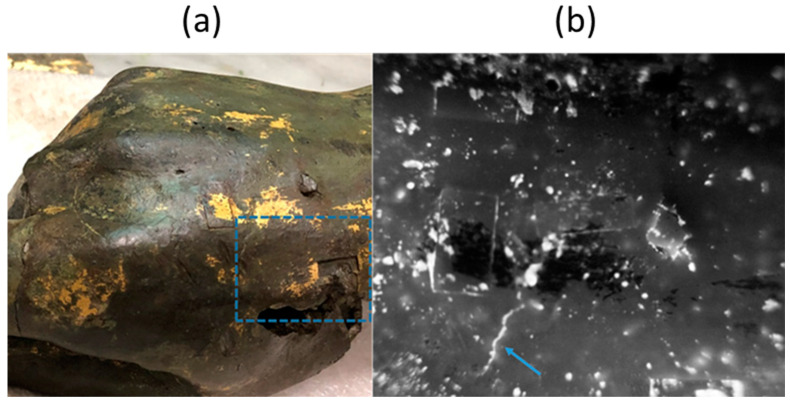
Crack under the visible surface of the hand (indicated by the blue arrow). Photograph (**a**) and thermogram (**b**) of the area marked by the dashed rectangle in (**a**) recorded at 0.016 s after the light flash.

**Figure 10 sensors-25-01727-f010:**
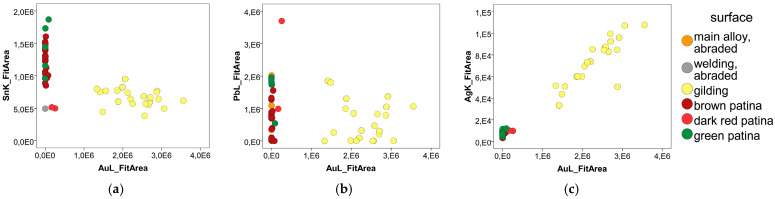
XRF: Peak area diagrams of Au-Sn (**a**), Au-Pb (**b**), and Au-Ag (**c**) for each surface type.

**Figure 11 sensors-25-01727-f011:**
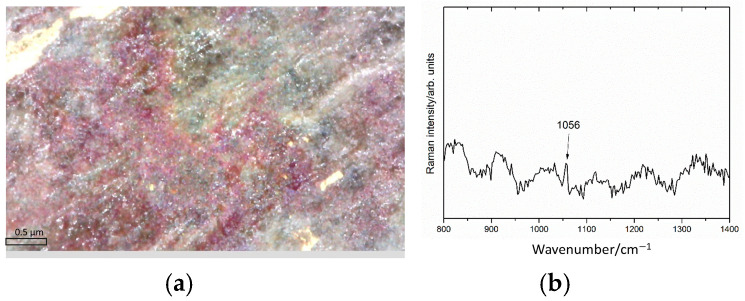
Detail of dark red patina (**a**) and its Raman spectrum (**b**).

**Figure 12 sensors-25-01727-f012:**
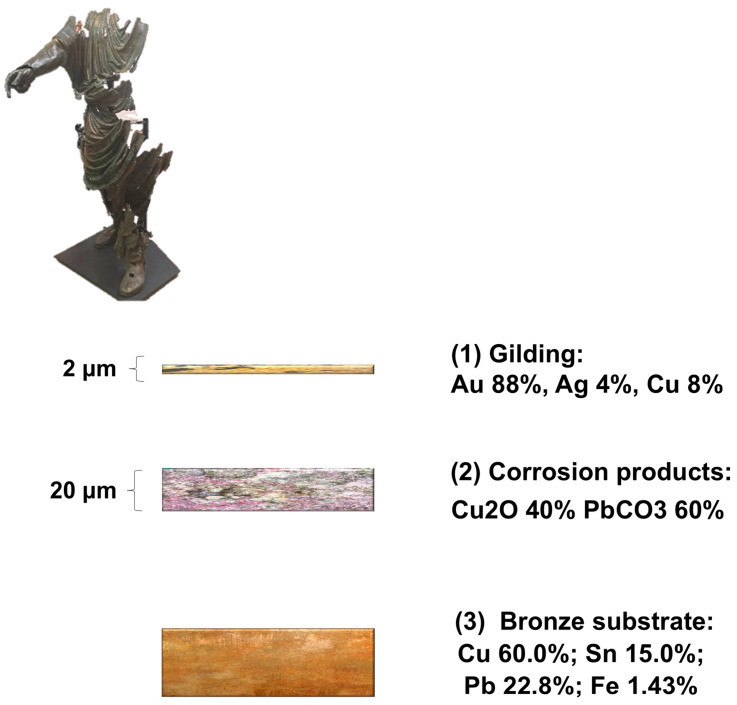
Simulation of surface stratification: average values of thickness and composition.

**Table 1 sensors-25-01727-t001:** Working conditions of the XRF spectrometer.

Anode	W
Voltage (kV)	70
Current (µA)	55
Filter	Cu 72 µm
Collimator	2 mm
Measurement time	100 s
Dead time	20%

**Table 2 sensors-25-01727-t002:** Model parameter values used for the numerical simulations. It is assumed that the insertion and the substrate are made of the same bronze type.

*H* [mm]	ρ [kg/m^3^]	*c* [J/(kg K)]	*k* [W/(m K)]	*I* [W/m^2^]
7	8800	380	56.85	1000

**Table 3 sensors-25-01727-t003:** Elemental composition of the Togato alloy and welding. The concentrations are normalised to 100.

	Mn%	Fe%	Co%	Ni%	Cu%	Zn%	As%	Ag%	Sn%	Sb%	Pb%	Bi%
**Alloy**	0.156	1.43	0.007	0.017	60.0	0.057	0.176	0.077	15.0	0.226	22.8	0.099
**Welding**	0.036	0.966	0	0	69.6	0	0.167	0.038	7.33	0.115	21.7	0.013

**Table 4 sensors-25-01727-t004:** Modelling performed with PyMca of the surface layers for each gilding spectrum of the Togato.

ID	Gilding Composition (%)			Gilding Thickness (µm)	Corrosion Layer Composition (%)		Corrosion Layer Thickness (µm)	Substrate Composition			
	Au%	Ag%	Cu%		Cu Oxide %	Pb Carbonate %		Fe%	Cu%	Sn%	Pb%
Tog004	86.26	3.82	9.92	3.0	52	48	18	1.44	60.47	15.13	22.96
Tog015	86.37	3.70	9.93	2.4	71	29	12.3				
Tog016	86.2	3.9	9.9	1.65	64	36	11				
Tog017	86.2	3.9	9.9	2.7	69	31	18				
Tog018	87.16	4.75	8.09	1.45	47	53	6				
Tog002	88.26	3.55	8.19	3.3	27	73	23				
Tog003	88.26	3.55	8.19	2.5	35	65	20				
Tog019	88.08	3.75	8.17	2.2	60	40	29				
Tog020	88.03	3.8	8.17	2.0	27	73	25				
Tog021	88.03	3.8	8.17	2.15	55	45	20				
Tog022	87.24	4.66	8.1	0.8	35	65	12				
Tog023	87.24	4.66	8.1	1.15	63	37	18				
Tog024	87.55	4.32	8.13	1.6	65	35	13				
Tog025	87.64	4.22	8.14	1.95	40	60	15				
Tog026	87.57	4.3	8.13	1.55	18	82	11				
Tog037	87.57	4.3	8.13	2.2	16	84	17				
Tog038	88.01	3.72	8.18	2.8	28	72	20				
Tog039	88.1	3.94	8.18	2.0	21	79	18				
Tog040	87.91	3.93	8.16	1.1	16	84	17				
Tog041	88.29	3.51	8.2	3.2	22	78	20				
Tog051	87.69	4.17	8.14	1.0	50	50	37				
Tog052	87.69	4.17	8.14	1.3	58	42	30				
Average Values	88	4	8	2	43	57	19				

## Data Availability

Data available on request.

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
