# Peer review of "Combined Use of Non-Destructive Analysis Techniques to Investigate Ancient Bronze Statues"

_sensors, 2025, doi:10.3390/s25061727_

Round 1
Reviewer 1 Report
Comments and Suggestions for Authors
The authors are kindly thanked for their interesting study, which presents a combined application of various NDT techniques on the ancient bronze statue Il Togato. The present work utilizes pulsed thermography with flash lamps, portable RAMAN spectroscopy, and XRF, offering the advantage of performing in-situ analyses. Additionally, numerical models were also incorporated to perform inverse analysis.
The paper is well presented; however, the methods section requires further integration to ensure the proposed procedure is fully replicable and to better explain some of the choices made in the study.
-
The details of the numerical model used should be clarified. The authors are kindly requested to specify the parameters and boundary conditions adopted in the model.
- What value was used for the input energy density?
- How was the energy distribution of the flash lamps modeled?
- How was the model validated?
- Were heat losses considered?
The authors are kindly asked to clarify these aspects and to include the parameters adopted in the model.
-
In Section 2.2.1, lines 170–183 describe general criteria that would be more appropriately placed in the introduction or in the theoretical background of pulsed thermography.
-
It is not clear whether multiple flash lamps were used, each with a nominal power of 3kW, or if this value refers to the total power. The authors are kindly requested to clarify the number of lamps used. Additionally, a schematic representation of the setup should be included, preferably as a complement to the existing setup photograph.
-
How was the non-uniformity of the surface emissivity handled? Could this have influenced the final results? The authors are kindly asked to provide clarification on this point.
-
Considering that the inspected surface, as shown in Figure 3, has a complex geometry, and thus the thermal flow cannot be assumed to be one-dimensional, can the assumptions made for the curves in Figure 4 still be considered valid?
-
The authors are kindly requested to verify the units of measurement in the temperature variation graphs over time. Although they are in a log-log scale, how can they start from values of 10710^7 °C? A check on the graphs in Figure 5 is also recommended.
-
The authors are kindly encouraged to emphasize the novelty of the proposed method and the combined approach, highlighting its advantages compared to individual techniques or more traditional methods.
These clarifications and improvements would strengthen the manuscript, ensuring greater reproducibility and a clearer understanding of the methodology and results.
Author Response
The authors wish to thank the anonymous Reviewer for his valuable comments on our paper, which helped us to improve the quality of our manuscript. All the Reviewer’s comments have been considered, and the manuscript has been changed accordingly. The changes in the revised version of the manuscript appear in red. A detailed list of the Reviewer’s comments together with our response are reported in what follows.
Reviewer
The authors are kindly thanked for their interesting study, which presents a combined application of various NDT techniques on the ancient bronze statue Il Togato. The present work utilizes pulsed thermography with flash lamps, portable RAMAN spectroscopy, and XRF, offering the advantage of performing in-situ analyses. Additionally, numerical models were also incorporated to perform inverse analysis.
The paper is well presented; however, the methods section requires further integration to ensure the proposed procedure is fully replicable and to better explain some of the choices made in the study.
- The details of the numerical model used should be clarified. The authors are kindly requested to specify the parameters and boundary conditions adopted in the model.
- What value was used for the input energy density?
- How was the energy distribution of the flash lamps modeled?
- How was the model validated?
- Were heat losses considered?
The authors are kindly asked to clarify these aspects and to include the parameters adopted in the model.
Response: The flash surface density used in the numerical simulations was 1 kW but, as reported in Table 2 added in the Result and Discussion section. However, its value is not essential for the comparison with experimental results since the mathematical model is linear and so the computed IR signal can be rescaled by multiplying it by a scalar quantity to match the corresponding experimental curve (a comment has been added in lines 347-351). The spatial density of the light pulse in the model is considered uniform on the illuminated surface of the specimen, as better clarified in line 176 of Section 2.1. A table (Table 2) has been added in the Result and Discussion section to specify the model parameter values assumed in the simulations. Heat losses can be easily accounted for in the model but have been neglected in these simulations since, in case of a bronze material who has a very high thermal diffusivity, their effect is negligible during the time course of the response (less than half a second). A comment on this has been added in lines 196-199 of the 2.1 Paragraph. Boundary conditions on the lateral and rear surface of the specimen is zero heat flux, whereas on the illuminated surface a flux equal to the lamp intensity is prescribed (see equation 2). The IR signal (which is essentially proportional to the surface temperature variations) in figures 4 and 5 is reported in arbitrary units. Accordingly, the units in the two figures have been changed from °C to a.u. The model has been validated in two previous papers both in the case of metallic materials (optically opaque) and in the case of semi-transparent materials (paper) by comparing the numerical results supplied by the model to experimental results obtained on test samples. To this end an explanatory paragraph has been added in Section 2.1 in lines 212-215 where the references of the two papers have been reported.
- In Section 2.2.1, lines 170–183 describe general criteria that would be more appropriately placed in the introduction or in the theoretical background of pulsed thermography.
Response: The lines have been placed in the introduction as suggested and all the corresponding paragraph has been suitably rewritten (lines 73-104).
- It is not clear whether multiple flash lamps were used, each with a nominal power of 3kW, or if this value refers to the total power. The authors are kindly requested to clarify the number of lamps used. Additionally, a schematic representation of the setup should be included, preferably as a complement to the existing setup photograph.
Response: Two flash lamps, each one of maximum power 3 kW, have been used in the experiments. This has been clarified in the Material and Methods section, lines 221-222. Moreover, a figure schematically depicting the experimental setup for the PT has been added in the Material and Methods section, as suggested by the reviewer (namely, Figure 3).
- How was the non-uniformity of the surface emissivity handled? Could this have influenced the final results? The authors are kindly asked to provide clarification on this point.
Response: The effect of a non-uniform emissivity of the bronze surface can be compensated by dividing pixel by pixel the signal in all the thermograms after the flash by the signal in the first thermogram just after the light pulse. This was not done in the experiments reported in the paper after having verified that the influence of a non-uniform emissivity was negligible for the scope of the present study. A comment on this issue has been added in the Materials and Methods, in lines 235-239.
- Considering that the inspected surface, as shown in Figure 3, has a complex geometry, and thus the thermal flow cannot be assumed to be one-dimensional, can the assumptions made for the curves in Figure 4 still be considered valid?
Response: As a matter of fact, the model is tri-dimensional and accounts for heat lateral diffusion, so the thermal flux inside the specimen is not necessarily one dimensional. Moreover, the patch dimensions are sufficiently small to be modeled as a flat surface. We don’t expect significant differences by modeling the specimen with a waved surface,
- The authors are kindly requested to verify the units of measurement in the temperature variation graphs over time. Although they are in a log-log scale, how can they start from values of 10710^7107 °C? A check on the graphs in Figure 5 is also recommended.
Response: The reviewer observation is correct. As reported in the response to point 1, the units in figure 4 and 5 on the vertical axis were wrong. The curves in the graphs are not a temperature variation but a IR signal (which is almost proportional to the surface temperature variation). The IR signal is reported in dimensionless unites since it is multiplied by a scaling constant to match the corresponding experimental results. The units have been accordingly changed in the two figures. A comment has been added in lines 347-351.
- The authors are kindly encouraged to emphasize the novelty of the proposed method and the combined approach, highlighting its advantages compared to individual techniques or more traditional methods.
Response: As suggested by the reviewer, the novelty of the combined approach and its advantages with respect to individual and traditional techniques has been better emphasized. A couple of paragraphs have been added in the Introduction (lines 34-55 and 73-82). More references have been added in the bibliography.
Reviewer 2 Report
Comments and Suggestions for Authors
The paper combines three non-destructive testing techniques—pulse thermography (PT), X-ray fluorescence (XRF), and Raman spectroscopy—to provide a comprehensive analysis of ancient bronze statues, offering important information about their manufacturing processes and conservation state. This multi-technique approach is innovative in the field of cultural heritage . However, there are the following issues in the paper:
1. The introduction does not sufficiently highlight the differences and unique contributions of this study compared to existing research. I suggest the authors further review the relevant literature, clearly identify the innovation of this paper, and discuss in more detail the practical significance of their findings for cultural heritage preservation and restoration.
2. The method section is detailed, especially regarding the mathematical model and experimental setup for pulse thermography. However, the descriptions of the experimental conditions for XRF and Raman spectroscopy are relatively brief, with particular omissions in the measurement parameters for Raman spectroscopy (such as integration time). I recommend the authors include these details to enable other researchers to replicate the experiments.
3. The data processing and analysis of XRF could be elaborated upon, particularly how the PyMCA software was used to construct and verify the layered model. The results section presents the experimental outcomes of the three techniques, especially highlighting the effectiveness of pulse thermography in detecting internal defects and repair traces on the bronze statues. However, the discussion section is somewhat brief and lacks an in-depth analysis of the results in conjunction with existing literature. I suggest the authors add a more detailed analysis and comparison with relevant studies in the discussion section.
4. The figures (e.g., thermography images, XRF spectra) are generally clear, but some of the labels and explanations are insufficiently detailed. For example, Figures 4 and 5 do not provide enough information about the selection of fitting parameters and error ranges. I suggest the authors add more explanatory text to the figures and include error analysis to improve the readability and reliability of the data.
5. The conclusion summarizes the main findings of the research, but it is brief and does not fully discuss the limitations of the study and potential directions for future research.
I recommend the authors perform a thorough language revision, particularly simplifying complex sentence structures to ensure clear and concise expression.
Author Response
The authors wish to thank the anonymous Reviewer for his valuable comments on our paper, which helped us to improve the quality of our manuscript. All the Reviewer’s comments have been considered, and the manuscript has been changed accordingly. The changes in the revised version of the manuscript appear in red. A detailed list of the Reviewer’s comments together with our response are reported in what follows.
Reviewer
The paper combines three non-destructive testing techniques—pulse thermography (PT), X-ray fluorescence (XRF), and Raman spectroscopy—to provide a comprehensive analysis of ancient bronze statues, offering important information about their manufacturing processes and conservation state. This multi-technique approach is innovative in the field of cultural heritage. However, there are the following issues in the paper:
- The introduction does not sufficiently highlight the differences and unique contributions of this study compared to existing research. I suggest the authors further review the relevant literature, clearly identify the innovation of this paper and discuss in more detail the practical significance of their findings for cultural heritage preservation and restoration.
Response: As suggested by the reviewer more discussion and comments on relevant existing literature have been added to the introduction. The novelty of the combined approach and its advantages with respect to individual and traditional techniques have been better highlighted (a couple of paragraphs have been added in lines 34-55 and 73-82). New references have been added to the bibliography.
- The method section is detailed, especially regarding the mathematical model and experimental setup for pulse thermography. However, the descriptions of the experimental conditions for XRF and Raman spectroscopy are relatively brief, with particular omissions in the measurement parameters for Raman spectroscopy (such as integration time). I recommend the authors include these details to enable other researchers to replicate the experiments.
Response: As suggested by the reviewer, more details have been added in the Materials and Methods section for Raman, specifically the missing parameters on the spectral acquisitions (lines 295-297) and for XRF the method used has been better detailed (lines 242-256). Details of the processing of the XRF data using PyMca software are also given in lines 273-281 to enable other researchers to replicate the experiments.
- The data processing and analysis of XRF could be elaborated upon, particularly how the PyMCA software was used to construct and verify the layered model. The results section presents the experimental outcomes of the three techniques, especially highlighting the effectiveness of pulse thermography in detecting internal defects and repair traces on the bronze statues. However, the discussion section is somewhat brief and lacks an in-depth analysis of the results in conjunction with existing literature. I suggest the authors add a more detailed analysis and comparison with relevant studies in the discussion section.
Response: as suggested by the reviewer, the discussion has been enriched with more data analysis and comparison with other existing studies. The processing and analysis of XRF data using PyMCA software to build and verify the 3-layer model is now has been well described in section 2.2.2 (line 241). The discussion is now included in the results chapter (renamed Results and discussion) in order to better explain the results and their interpretation in the light of knowledge of large bronze statuary and the different gilding techniques (lines 406-414 and 425-427).
- The figures (e.g., thermography images, XRF spectra) are generally clear, but some of the labels and explanations are insufficiently detailed. For example, Figures 4 and 5 do not provide enough information about the selection of fitting parameters and error ranges. I suggest the authors add more explanatory text to the figures and include error analysis to improve the readability and reliability of the data.
Response: The caption of figures 4 and 5 and the relevant explanations in the Results section have been enriched with more details concerning fitting parameters and error ranges.
- The conclusion summarizes the main findings of the research, but it is brief and does not fully discuss the limitations of the study and potential directions for future research
Response: As recommended by the reviewer, the conclusions section has been revised, further emphasizing the main results and better explaining the limitations of the research. Regarding planning future research, based on the authors' previous studies on large bronze statues and thanks to the results of the present investigations, which are completely non-invasive, it is intended to continue the study of the gilding of ancient bronzes by selecting other statues in order to characterise the different gilding techniques used in the ancient world to embellish monumental statues. Future research will be based on the one hand, on the implementation of a database of the thermographic features of large bronzes (technical peculiarities observed with the IRT) and on the other hand on the application of modelling for the study of gilding through the processing of XRF data (lines 511-517).
Round 2
Reviewer 1 Report
Comments and Suggestions for Authors
Now the paper is suitable for publication
Reviewer 2 Report
Comments and Suggestions for Authors
This manuscript has been revised based on review’s comment. I think the manuscript can be accepted.